# Current Landscape of Sonodynamic Therapy for Treating Cancer

**DOI:** 10.3390/cancers13246184

**Published:** 2021-12-08

**Authors:** Toshihiro Yamaguchi, Shuji Kitahara, Kaori Kusuda, Jun Okamoto, Yuki Horise, Ken Masamune, Yoshihiro Muragaki

**Affiliations:** Faculty of Advanced Techno Surgery (FATS), Institute of Advanced Biomedical Engineering and Science, Tokyo Women’s Medical University, Shinjuku, Tokyo 162-8666, Japan; yamaguchi.toshihiro@twmu.ac.jp (T.Y.); Kitahara.shuji@twmu.ac.jp (S.K.); kusuda.kaori@twmu.ac.jp (K.K.); jun_okamoto@sonire-t.com (J.O.); horise.yuki@twmu.ac.jp (Y.H.); masamune.ken@twmu.ac.jp (K.M.)

**Keywords:** SDT, HIFU, DDS, cavitation, prostate cancer, glioma, PDAC

## Abstract

**Simple Summary:**

Recently, ultrasound has advanced in its treatment opportunities. One example is sonodynamic therapy, a minimally invasive anti-cancer therapy involving a chemical sonosensitizer and focused ultrasound. The combination of the ultrasound and chemical sonosensitizer amplifies the drug’s ability to target cancer cells. Combining multiple chemical sonosensitizers with ultrasound can create a synergistic effect that could effectively disrupt tumorigenic growth, induce cell death, and elicit an immune response. This review provides an oversight of the application of this treatment to various types of cancer, including prostate cancer, glioma, and pancreatic ductal adenocarcinoma tumors.

**Abstract:**

Recent advancements have tangibly changed the cancer treatment landscape. However, curative therapy for this dreadful disease remains an unmet need. Sonodynamic therapy (SDT) is a minimally invasive anti-cancer therapy involving a chemical sonosensitizer and focused ultrasound. A high-intensity focused ultrasound (HIFU) beam is used to destroy or denature targeted cancer tissues. Some SDTs are based on unfocused ultrasound (US). In some SDTs, HIFU is combined with a drug, known as a chemical sonosensitizer, to amplify the drug’s ability to damage cancer cells preferentially. The mechanism by which US interferes with cancer cell function is further amplified by applying acoustic sensitizers. Combining multiple chemical sonosensitizers with US creates a substantial synergistic effect that could effectively disrupt tumorigenic growth, induce cell death, and elicit an immune response. Therefore, the minimally invasive SDT treatment is currently attracting attention. It can be combined with targeted therapy (double-targeting cancer therapy) and immunotherapy in the future and is expected to be a boon for treating previously incurable cancers. In this paper, we will consider the current state of this therapy and discuss parts of our research.

## 1. Introduction

The environment surrounding cancer treatment has changed rapidly in recent years; however, an actual cure for this terrible disease has yet to be found. Sonodynamic therapy (SDT) can be a non-invasive method of anti-cancer treatment using chemical sonosensitizers and high-intensity focused ultrasound (HIFU). It works by using a concentrated, high-intensity ultrasound (US) beam to destroy or denature the targeted cancerous tissue. By combining US with a medication called a chemical sonosensitizer, we amplify the drug’s ability to damage cancer cells preferentially. This result is based on the theory that US can lead to a substantial physical and chemical change in cancer cells. The mechanism by which US inhibits cancer cell function is further amplified by applying an acoustic sensitizing drug. Combining various chemical sonosensitizers with US creates a substantial synergistic effect that can inhibit neoplastic growth, induce cell death, and trigger an immune response.

For this reason, US therapy has been studied for various therapeutic applications, including HIFU, in which US waves are focused by an acoustic lens to cauterize the affected areas. In SDT, an acoustically sensitive substance is irradiated and activated by relatively low intensity US waves to exert a cell-killing effect. Sonoporation is considered an effective treatment for improving drug transfer into cells, while with gene transfer, genes are delivered without cell damage these various therapeutic applications are being studied (Table 1). In the future, US therapy may be used in combination with targeted therapy (double-targeting cancer therapy) and immunotherapy. 

In this paper, we considered the current status and aspects of research in this area, including the history of US treatment, cavitation action, HIFU, SDT, and explain sonosensitization. Furthermore, we discuss the US treatment of various types of cancers, including prostate cancer, glioma, and pancreatic ductal adenocarcinoma tumors.

## 2. Ultrasound Treatment

### 2.1. The History of Ultrasound Treatment for Cancer

In general, many combined treatments show a greater synergistic effect than the sum of the effects of two treatments conducted separately, which is not limited to the US field only. US was previously used for hyperthermia, and was then developed into microbubble formulation, drug delivery systems (DDS), HIFU, focused ultrasound, sonoporation, and gene transfer. Recently, clinical applications such as magnetic resonance imaging-guided focused ultrasound (MRgFUS), low-intensity pulsed ultrasound (LIPUS), and the blood-brain barrier (BBB) opening, have developed rapidly.

Since the late 1980s, US-based therapies have been studied, such as with hyperthermia and SDT. SDT uses drugs that concentrate on a specific area, such as a tumor, and increase its sensitivity to US, in combination with US irradiation. In the late 1980s, it was reported that a variety of photosensitizing agents and US had a high anti-tumor effect, and many US-sensitizing agents, known as SDT, were studied and compared to photodynamic therapy (PDT). SDT is still being investigated but is expected to be a profound cancer treatment modality. From the end of the 1980s to the 1990s, Umemura et al. researched various chemical substances. They found that porphyrin-based substances exhibit anti-tumor effects and are activated by acoustic cavitation produced by US irradiation; thus, they suggested this therapy.

Currently, US is widely used for tumor diagnosis and treatment, and HIFU and SDT are considered promising non-invasive therapies in cancer treatment [1]. One of the most popular methods is SDT, a less invasive treatment for solid tumors using US and chemicals (ultrasound sensitizers) [2].

### 2.2. Cavitation

The phenomenon known as cavitation has been recognized as a major mechanism of SDT. The reactive oxygen species (ROS) created by the US are chemically active components containing oxygen. When these ROS are generated at high levels in the biological body, oxidative stress occurs at various levels in the cells [3,4,5,6,7]. Cavitation is defined as the mechanical force exerted by sound waves on microbubbles in a fluid. As the sound waves are propagated through the fluid, the characteristic compression and rarefaction causes the microbubbles in the tissue fluid to contract and expand [8].

There are two forms of cavitation: stable and inertial cavitation. Stable cavitation is the stable oscillation of the bubble size when exposed to a low-pressure sound field while inertial cavitation refers to the violent oscillation of bubbles and their rapid growth during the dilatation phase when they reach their resonant size, eventually leading to a violent collapse and destruction of the bubbles. When a violent collapse occurs, a high-pressure, high-temperature shock wave is generated in the microenvironment. The rapid movement of fluid in the vicinity of the bubble due to the oscillatory motion of stable cavitation is called microstreaming. Microstreaming can generate high shearing forces that can cause transient damage to cell membranes; thus, it may enable the delivery of drugs and genes using US [8,9,10]. However, it is challenging to monitor cavitation effectively, and there is a risk of relatively serious side effects if it is not applied correctly.

### 2.3. Chemical Sonosensitizers and Focused Ultrasound

The clinical application of HIFU is being actively studied. Currently, HIFU irradiation is being used to treat tumors of the prostate, bone, liver, pancreas, stomach, brain, uterus, kidney, thyroid, and other organs. There are two mechanisms of HIFU ablation: thermal and mechanical effect. The thermal effect is the generation of heat due to the absorption of acoustic energy, accompanied by a rapid increase in temperature in the local tissues. When the tissue temperature rises higher than 60 °C, coagulation necrosis occurs, resulting in tumor cell destruction [11,12]. On the other hand, mechanical effects are only associated with high-intensity acoustic pulses, including cavitation, microstreaming, and radiation forces [8]. Although thermal damage to irradiated and surrounding tissues is a concern during HIFU irradiation, thermal damage to tissues during high-temperature exposure is almost linearly dependent on the exposure time, and exponentially dependent on temperature rise [13].

In recent years, MRgFUS, a non-invasive thermal ablation method, has been developed as an integrated system with HIFU treatment. The US-guided method is suitable for preoperative tumor localization, but not for evaluating the treatment boundaries intraoperatively, due to the acoustic contrast generated by the thermal tissue bubbles. Conversely, the MRI-guided method is suitable for transient tissue temperature measurement, although it cannot effectively measure lethal calorific value [14]. For this reason, the development of thermally sensitive microbubbles is being considered to enhance US imaging of the lethal heat content [15]. Histotripsy mechanically destroys tissues using high-intensity pulsed US. This technique has been attracting attention as a treatment method using non-thermal effects and has the following advantages compared to HIFU: less susceptibility to the effects of blood flow; easy monitoring by US; and advantages over the MRI-guided method mainly in terms of cost and ease of application [16].

### 2.4. Sonodynamic Therapy

SDT is a treatment that utilizes the synergistic effects of US and chemical compounds. In SDT, US stimulation, especially cavitation, causes cytotoxicity, making it one of the least invasive treatments for solid tumors [2,7,17]. SDT depends on the simultaneous combination of low-intensity US, molecular oxygen, and US sensitizers to generate ROS. This approach is similar to that of PDT, which uses light instead of US to activate the sensitizers [5,18]. One of the advantages of SDT is that US reaches deeper into the tissues compared to how far PDT can penetrate [18], which allows SDT to be used instead of PDT, which uses less penetrating light and produces cytotoxic effects on sensitized tissues. This suggests that SDT may be applied more widely to tumor cases and clinically applied as a non-invasive treatment for solid tumors that were previously too deep to access [18,19]. Of the extensive in vitro and in vivo research, US parameters in cancer treatment using SDT are the most important ones [17]. Sonoluminescence is a phenomenon in which the energy gained from the rapid collapse of bubbles during acoustic cavitation causes a very short period of luminescence. Sonoluminescence is generally associated with inertial cavitation due to the rapid release of energy from a bubble collapse. However, sonoluminescence has also been observed at acoustic pressures and amplitudes resulting from stable cavitation. Although the possibility of such a phenomenon has hardly been investigated, if sonoluminescence occurs under stable cavitation conditions, it is expected to provide more control over the activation of the sonosensitizer and enhance the synergy between ultrasound and sonosensitizer, without the influence of inertial cavitation [20]. Additionally, recent developments in micro/nanotechnology have encouraged the use of US in biomedicine, especially in the field of DDS, which is a system for transporting therapeutics in the body as needed in order to achieve safe and efficient desired therapeutic effects and oncology. US activates DDS through various mechanisms and enables US-triggered drug release that targets only tumor sites [21]. There are many combinations between available US sensitizers, each tumor model, and the US parameters for activating these sensitizers. Many US sensitizers preferentially accumulate in tumor regions but are also located in other tissues. Since many of the sensitizers used in PDT and SDT are the same, problems in PDT, such as hypersensitivity to light, suggest similar issues for SDT. Therefore, new targeted drug delivery strategies that allow tumor-specific delivery of ultrasensitizers will be more effective for SDT [7].

There are various mechanisms by which SDT enhances anti-tumor activity in promoting the change from M2 to M1 macrophages in the tumors [22]. M1 macrophages are responsible for the inflammatory process by secreting pro-inflammatory cytokines, and they play a role in anti-tumor activity. On the other hand, M2 microglia/macrophages suppress the inflammatory response and play a significant role in tissue repair. Malignant gliomas are infiltrated by M2 microglia/macrophages, which promote immunosuppression and are involved in tumor progression. It has been reported that mice subjected to 5-aminolevulinic acid (5-ALA) SDT have a higher number of M1 CD68+ macrophages and a markedly reduced number of M2 CD163+ macrophages [23,24,25,26]. It has also been reported that the expression levels of CD68 and CD80, markers of dendritic cell maturation in vivo, were significantly higher in SDT-treated mice, suggesting that SDT may promote dendritic cell maturation and enhance anti-tumor immunity [23]. SDT may enhance immunogenicity by targeting the tumor microenvironment and tumor cells, acting on infiltrating immune and tumor cells.

SDT has shown superior efficacy in the past, suggesting that it may treat cancer in tumors with no currently-adequate treatment. Additionally, when combined with other therapies (e.g., chemotherapy, immunotherapy, etc.), it may be a less invasive treatment.

### 2.5. Sonosensitizer

Conventional sonosensitizers have been developed based on specific types of organic molecules and fall into four major categories: porphyrins, phthalocyanines (Pcs), xanthenes, and anti-tumor agents [27]. The categorization of sonosensitizers is shown in Table 2.

## 3. Cancer-Specific Treatments

### 3.1. Prostate Cancer

Prostate cancer is the most common malignant neoplasm in the male population [28,29]. As male life expectancy has increased over the past 25 years, the age at which prostate cancer is detected has decreased by an average of 10 years [30]. This pattern indicates the limitations of conventional treatments for prostate cancer, including the risk of recurrence and long-term morbidity from urinary system diseases [31,32].

Currently, there is a wide range of treatment options for prostate cancer, depending on the severity of the disease. For low and medium-risk prostate cancer, options include active surveillance, minimally invasive resection therapy, radiation therapy, and prostatectomy [33]. In addition, radiation therapy, brachytherapy, and prostatectomy are recommended for localized cancers [34,35]. In contrast, chemotherapy-induced recurrence is common, and chemotherapy often induces serious toxic effects [36].

Two different treatments (HIFU and LIPUS), based on the US field, are currently available. Studies on the use of HIFU have been conducted since the 1990s. Compared to other treatment methods, HIFU ablation has the advantage of not causing substantial tissue damage outside the treatment areas [37]. The significant advantage of HIFU is that it is a non-invasive treatment and does not require the insertion of a probe into the target tissue [38]. HIFU is considered superior to other methods because it has lower post-treatment incidences than other techniques [39,40].

In contrast, LIPUS causes stable cavitation without increasing the temperature. LIPUS has a lower intensity compared to HIFU, so the thermal effects are reduced. Most of the impact of LIPUS will be mechanical or involve non-thermal cellular changes. LIPUS is being investigated as a treatment for prostate cancer, either alone or in combination with either microbubbles or anti-cancer drugs. Microbubbles cause cavitation, which results in shear stress and permeability incell membranes, allowing the drug to enter the cells and strengthen its anti-cancer effects. The advantage of this technique is in its ability to focus the US energy on targeted tissues to induce local cytotoxicity by activating sonosensitizers with minimal damage to healthy tissues [41]. There is also a need for improved measures so that drug-resistant cancers can be treated with chemotherapy. LIPUS, in combination with anti-cancer drugs, could provide a new treatment for drug-resistant cancers [41]. LIPUS can penetrate deeper than HIFU and thus has a broader range of clinical applications. This is the reason why US at higher frequencies affects tissues that are more superficial; at a lower frequency, less energy is absorbed superficially and more is available to penetrate into deeper tissues [42]. In addition, LIPUS causes less damage to cells and is safer for normal tissues.

While many previous studies have validated the efficacy of SDT as a single treatment, recent years have seen the utilization of an approach of repeated SDT treatments with superior results [43]. In the case of prostate cancer, repeated SDT treatments, using either extracorporeal or transrectal US transducers, can be easily applied in a clinical setting, is non-invasive, and has the potential to eliminate the tumor with minimal side effects. SDT is likely to become the first-line treatment for prostate cancer patients, especially those who do not meet the eligibility criteria for standard local ablation methods, such as HIFU. Since SDT does not require the insertion of electrodes or probes inside or near the affected area, the treatment can be performed with minimal effects on the tissue [44].

### 3.2. Glioma

Malignant glial tumors are the second most common reason for death from central nervous system diseases, second only to stroke. Glioblastoma multiforme (GBM) is the most common glioma (about 50% of cases) and is one of the most aggressive malignancies occurring in adult patients. Currently, a combination of surgery, radiotherapy, and chemotherapy is used. Still, no matter what treatment is used, the median survival time for GBM is 1–1.5 years, and the five-year survival rate is less than 5% [45,46,47]. The inadequate response to treatment of GBM is attributed to the self-renewal of rapidly proliferating tumor stem cells (TSCs), which are resistant to chemotherapy and radiotherapy. TSCs infiltrate healthy brain tissue, and their axonal pathways are far removed from the glioma lesion, leading to their recurrence [48].

One of the features that explains the poor efficacy of chemotherapy for malignant gliomas is the BBB, which blocks the effects of many drugs in the central nervous system, limiting the range of effective chemotherapeutic drugs [49]. Even if enough cytotoxic drug is administered to the part of the tumor where the BBB has been destroyed, the drug concentration is thought to be several times lower around the tumor where TSCs surrounded by the BBB are present. These characteristics of malignant glioma result in tumor recurrence, regardless of whether the lesion has been surgically removed entirely or not. As current therapies have not yielded satisfactory results, new therapeutic approaches, including immunotherapy strategies, are being investigated to inhibit tumor progression, eradicate invasive neoplastic cells, and treat unresectable masses [50].

Considering the ability of US to penetrate tissue and accumulate acoustic energy in a small volume within the tissue, SDT could be an excellent treatment, especially for hard-to-reach and deeply localized tumors such as glioma. Furthermore, if the US sensitizer selectively accumulates in cancer cells, SDT could be a safer method that would not harm healthy brain cells [51]. Moreover, reviews on SDT and glioma discuss various potential applications of SDT, including BBB-opening, increased drug delivery to tumor cells, and enhanced immunotherapy [52,53]. However, the mechanism by which SDT exerts its cytotoxic effect on brain tumors is not well understood. The most plausible theories include the cavitation effect, the generation of ROS, the induction of apoptosis, the enhancement of anti-tumor immunity, the suppression of angiogenesis, and the induction of hyperthermia [54].

Recently, SDT has been considered as a new approach for high-grade glial neoplasms and an alternative treatment for unresectable masses, by enhancing standard treatments for delaying tumor recurrence. Indeed, by taking advantage of the specific pharmacokinetics of 5-ALA and fluorescein sodium, SDT can selectively narrow the cytotoxic and modulatory effects on glioma cells while sparing the surrounding parenchyma. This concept is important in the field of neuro-oncology, where neural tissue near the tumor lesion may be involved in many functions and should be preserved to maintain the patient’s quality of life [22].

US sensitizers have been studied, especially 5-ALA and fluorescein, which are already widely used to guide the resection of malignant brain tumors due to their selective accumulation in glial cells and known good safety properties. These characteristics make them good candidates for experimental studies in SDT [55,56]. 5-ALA will be the most widely employed porphyrin-based sonosensitizer in the in vivo glioma model of SDT. 5-ALA, an amino acid precursor required for heme biosynthesis, is produced in mitochondria from glycine and succinyl-CoA by 5-ALA synthase. Protoporphyrin IX (PpIX), the final metabolite of the heme biosynthetic pathway, has photosensitizing properties. The administration of 5-ALA results in increased synthesis of PpIX, which accumulates in cancer cells, mainly in the brains of patients with glioma [57]. The light active PpIX emits red light at wavelengths of 635 and 704 nm when excited by blue light at 380-420 nm wavelengths. This property of protoporphyrin has been exploited in neurosurgery for fluorescence-guided resection procedures [57,58,59]. However, because gliomas are invasive, it is difficult to remove cancer cells altogether, and the low accumulation efficiency of 5-ALA in glioma cancer stem cells prevents accurate resection.

The new technique of using sonosensitizer and US for glioma treatment is selective for malignant cells, so there is no need to identify them in advance. At the same time, it is non-invasive and can be applied repeatedly. The availability of SDT with 5-ALA has been evaluated in a rat model of glioma C6. When healthy brain tissue was irradiated with US at a frequency of 1.04 MHz and a local intensity of 15 W/cm^2^ for 5 min, a large amount of brain tissue was found to be lost. Therefore, SDT used US of lower intensity (10 W/cm^2^) and the same frequency and duration of irradiation. A significant reduction in tumor size was observed in the brains of animals treated with SDT compared to either the control group or the group irradiated only with US [60]. In an experiment using a rat model of glioblastoma C6 transplanted into Sprague Dawley rats, favorable treatment response and increased survival were observed in the group that received SDT via 5-ALA. This result was not observed in the 5-ALA-only or US-only groups [61].

Although many studies in the past have used frequencies of around 1 MHz, 5-ALA-SDT using frequencies as low as 25 kHz showed tumor regression and growth inhibition in an in vivo U87-MG glioma model. This frequency is used in many surgical US aspirators, and the authors advocate the intraoperative application of SDT using such devices in combination with 5-ALA [62]. On the other hand, fluorescein, an organic compound belonging to the xanthene family of dyes, is a suitable compound for the resection of malignant gliomas in neurosurgery due to its selective accumulation in the brain regions where the BBB is impaired, in addition to its rapid flushing from blood vessels and normal tissues [55,63]. In a study using a rat-C6 glioma model, fluorescein-based SDT (FL-SDT) not only showed selective accumulation of the compound in subcutaneously injected tumors, reaching a peak at 30′, but FL-SDT also showed efficacy compared to either a control group or a group that received US without sensitizers.

While the latter group only experienced delayed growth due to treatment, FL-SDT showed a mild reduction in tumor volume at the seven-day checkpoint. The study did not show any significant apoptotic markers or DNA fragmentation trends, but this was probably due to the delayed checkpoint [64,65]. Fluorescein and 5-ALA, which have been investigated in various preclinical studies using glioma models, are widely used in neurosurgery to induce resection of malignant brain tumors. These compounds are known to be safe and selectively accumulate in glial cells in vivo and should be the focus of attention for further research and clinical application [22]

### 3.3. Pancreatic Duct Adenocarcinoma (PDAC)

PDAC has a poor prognosis among gastrointestinal tumors, with no specific early symptoms, and many tumors are inoperable at diagnosis. The median survival time is only four to six months, and the five-year survival rate without treatment is less than 1%. Newer chemotherapies have been introduced yet the 1-year survival rate is still only about 20% [66].

Chemotherapy has a limited effect on local tumor control and pain and symptom reduction. As a result, many affected patients will have their quality of life significantly impacted. The purpose of local therapy for pancreatic cancer is to reduce complications associated with the tumor, and to relieve symptoms. Currently, radiation therapy has been established as a local therapy, but it merely prolongs the survival period and alleviates symptoms in clinical terms. In recent years, other local ablation methods such as cryotherapy, radiofrequency ablation, microwave ablation, irreversible electroporation, and HIFU have been used with good results in some cases [67]. In addition to chemotherapy, minimally invasive ablative therapy is available for unresectable PDAC, where radical surgery is impossible, and chemotherapy has limited efficacy. Since the late 1990s, HIFU has been used and is now recommended as an alternative treatment for unresectable PDAC [68,69]. HIFU effectively resects pancreatic tumors by increasing the local tissue temperature to 65 °C, breaking down tumor cells, breaching the pancreatic cancer stromal barrier, and facilitating chemotherapy delivery to the pancreatic tumor [70]. Several studies have reported that HIFU combined with chemotherapy has shown better results than chemotherapy alone.

In a recently developed genetically engineered mouse model, mutated alleles of Kras and p53 are expressed in pancreatic cells, resulting in tumors that closely resemble the pathophysiology and molecular characteristics of human PDAC [71,72]. As such, this animal provides a more realistic model for evaluating the potential for future therapies, especially drug delivery [73]. The efficacy of HIFU-induced hyperthermia combined with low-temperature liposomes to enhance doxorubicin delivery was evaluated in the PDAC model using the KPC mouse model. In the study results, targeted hyperthermia with FUS was performed after systemic administration of doxorubicin-filled cold-sensitive liposomes (LTSL-Dox). Monitoring using MR-thermometry showed a twofold increase in the median amount of doxorubicin accumulation in the target tumor tissue compared to the same amount of doxorubicin administered without encapsulation. There are two possible mechanisms for the increased drug accumulation, vascular changes due to hyperthermia and local drug release. Mild hyperthermia (40–43 °C) has been shown to increase tumor blood flow and vascular permeability [74,75,76,77]. The increase in drug concentration when hyperthermia is combined with LTSL drugs can be attributed to the changes in tumor vascular properties caused by hyperthermia and the high local concentrations of available medications released in the vascular system [78]. In this research, continuous-wave HIFU was successfully used to provide mild heat to enhance drug delivery.

In addition, other biological effects of HIFU, such as cavitation, can be used to enhance drug penetration. In a previous study, pulsed HIFU, to disrupt the stroma, increased the permeability of the pancreatic tumor stroma and enhanced drug penetration [70]. This suggested that the combination of mild hyperthermia with HIFU and mechanical disruption may further enhance drug penetration [73]. Currently, gemcitabine is the standard treatment for pancreatic cancer, and doxorubicin is not used to treat pancreatic cancer. However, many studies are evaluating new ways of delivering doxorubicin to improve local accumulation without systemic toxicity, which may lead to the use of doxorubicin for pancreatic cancer in the future [79,80,81].

## 4. Conclusions

Cancer treatment is now undergoing remarkable changes, and new reports on topics such as HIFU, SDT, and DDS are increasing yearly. There have also been significant developments in the clinical aspects of handling tumors, such as attempts to use US to improve the BBB-opening, LIPUS to suppress cancer cells, the development of nano/microtechnology for therapeutic applications, and the clinical application of MRgFUS. However, there are still few clinical reports and many issues to be solved, such as the analysis of the detailed mechanism of US therapy. Our research team has also conducted SDT experiments using NC-6300 in the past.

We conducted clinical experiments using human pancreatic adenocarcinoma cells (BxPC-3), a mouse colorectal cancer cell line (colon-26), and human pancreatic cancer cells (MIA Paca-2) as mouse tumor models, as well as canine chondrosarcoma, osteosarcoma, and hepatocellular carcinoma [82,83,84]. The mechanism of US’s biological effects is very complex, and it is important to know how to apply the necessary effects to the target area of treatment. Therefore, it is necessary to conduct further research on US therapy for cancer treatment and to develop new and additional US control technologies. These new therapies, which have the potential to improve current treatment strategies while maintaining the quality of life of cancer patients, are expected to be used in combination with other therapies such as targeted therapy and immunotherapy in the future and are expected to become a new therapeutic pillar in the treatment of cancers which were previously considered incurable (Table 3).

## Figures and Tables

**Table 1 cancers-13-06184-t001:** Comparison of ultrasound therapy.

	HIFU	SDT
**Mechanism of action**	Cauterization of the affected area by focused ultrasound	Activation of sonosensitizer by ultrasound
**Therapeutic value**	Tissue necrosis and thermal coagulation due to temperature increase	ROS generation, mechanical stress

	**Sonoporation**	**Gene transfer**
**Mechanism of action**	Transient small pore formation due to microbubble collapse	Gene transfer by sonoporation
**Therapeutic value**	Introduction of various substances into the cell	Drug delivery, non-viral gene transfer


**Table 2 cancers-13-06184-t002:** Categorization of sonosensitizer.

Category		Sonosensitizers	
Porphyrins		HP,PplX,HMME,DVDMS	
Phthalocyanines(Pcs)		ZnPcS_2_P_2_,AlPcS_2_a	
Xanthenes		Er,RBD2,RBD3	
Antitumor drugs		Adriamycin,Artemisinin	
Anti-inflammatory drugs		Piroxicam,Levofloxacin	
Others		5-ALA,HB,ICG,PHF,Ce6	

Porphyrins: Porphyrins are considered to be one of the first generation photosensitizers. Pcs: Phthalocyanine is a second-generation photosensitizer. Xanthenes: Xanthenes are dyes. Antitumor drugs: A number of anti-tumor agents can be used as sonosensitizers. Anti-inflammatory drugs: Nonsteroidal anti-inflammatory drugs show anti-tumor effects. Others: Other organic molecules can also activate SDT. HP: hematoporphyrin; PpIX: protoporphyrin; HMME: hematoporphyrin monomethylether; DVDMS: sinoporphyrin sodium; Er: erythrosine B; RBD (RBD2, RBD3): Rose bengal derivative 2,3; 5-ALA: 5-aminolevulinic acid; HB: hypocrellin B; ICG: indocyanine green; PHF: polyhydroxy fullerenes; Ce6: Chlorin e6.

**Table 3 cancers-13-06184-t003:** Development and prospects of cancer therapy using ultrasound.

Present	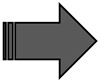	Future
US (+ Radiation/Chemotherapy)		The analysis of the detailed mechanism of US therapy	
US + Microbubble/Nanobubble		To develop new and further US control technologies	
Nanoparticle combination		US + Targeted therapy		
Gene transfer using US		US + Immunotherapy		
BBB-opening			Development of new sonosensitizer and SDT	
HIFU/LIPUS						
DDS/SDT						
MRgFUS						
Histotripsy

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
