# Peer review of "Current Landscape of Sonodynamic Therapy for Treating Cancer"

_cancers, 2021, doi:10.3390/cancers13246184_

Round 1

Reviewer 1 Report

This manuscript discusses an emerging therapy against cancer called sonodynamic therapy, SDT, and, although this review seems very interesting, before the publication the authors should discuss in more details some aspects that in my opinion are poorly described or missing. In this regard I suggest first, to discuss more deeply the mechanism behind sonodynamic therapy and also introducing the intriguing idea about sonoluminescence.

Therefore, please refer to these papers:

DOI: 10.1021/acs.bioconjchem.0c00029; DOI: 10.3390/cancers13153852;

DOI: 10.1039/D0PP00133C;

DOI: 10.1016/j.freeradbiomed.2018.05.002;

DOI: 10.1121/1.3675003.

Moreover, if this review wants to be comprehensive about this topic, I think that all tumors, where this therapy has been investigated, must be reported and not just some of them.

Therefore, I suggest:

  • introducing a comprehensive table reporting all the studies where the sonodynamic has been involved in vitro and in vivo with a short comment about the results;
  • analyze the value of the US field parameters, pressure and frequency, used in the SDT therapies.

I suggest looking more deeply about the breast cancer where some interesting papers have been already published in this field such as:

DOI: 10.3390/ph14100972;

DOI: 10.2217/nnm-2016-0293;

DOI: 10.2217/nnm.15.150.

Line 9: Ultrasound has been a modestly effective treatment for cancers over recent years.

> Not really true, strongly depends on the type of cancer. For example there are good results for the treatments of prostatic cancer

Please delete or change the statement

Line 20: A focused high-intensity convergent ultrasound (US) beam is used to destroy or

> A high-intensity focused ultrasound (HIFU) beam is used to destroy

Focused means convergent

Line 21: This US is combined with a drug, known as a chemical

> In some type of SDT therapy HIFU is combined with a drug, known as a chemical

There are some SDT therapy based on unfocused ultrasound. Please, consider them.

Line 22-23: This result is based on the theory that US can exhibit substantial physical and chemical changes in cancer cells

> This is not completely true, it is not well known the mechanism of combined action of ultrasound and sonosensitizer.

Please delete or change the statement.

 Line 34: 1. Introduction

>Please see and use the above suggestions.

Line 58-59: The history of cancer treatment using US dates back to the early days when US alone

> Also today there are many cancer treatment based only on US.

Please change the statement.

In general there are many combined treatments that show a greater synergistic effect than the sum of the effects of the two treatments conducted separately, not only in US field.

Line 69: called SDT

> SDT is already defined

Line 44: Combining several chemical sonosensitizers

> several is too generic (like multiple), please add a value i.e. 2,3

Line 80: US breaks down its propagating medium and generates bubbles

> US doesn’t generate bubbles, bubbles are in the medium.

Please delete or change the statement

Line 100: not applied correctly

> not correctly controlled.

Line 185: There are two types of US intensities used for therapeutic purposes:

> Two different treatments, based on ultrasound field, are currently available:

Line 289-290: When healthy brain tissue was irradiated with US 289 at a frequency of 1.04 MHz and local intensity of 15 W/cm2 for 5 minutes

> In continuous mode? If not please specify Pulse Repetition Frequency, PRF.

Line 290-291: it was found 290 that a large amount of brain tissue was being lost

> Please specify the percentage

Reviewer 2 Report

The article describes the history of sonodynamic therapy (SDT), mechanism of action and components of SDT for cancer treatment, and SDT applications of three cancers. The configuration is well defined, but it's lack of a deep insight into the current challenges of SDT in terms of its technique and molecular mechanism. Here are some other comments for better readability.

  1. Please discuss what is the primary differences in mechanism of action and efficacy of cancer treatment between HIFU, SDT, sonoporation and gene transfer, and list a summary table.
  2. It's unclear which one of stable cavitation and inertial cavitation dominates SDT. Please clarify it.
  3. Ultrasonic parameters such as the operating frequency, pulse width, duty cycle, spatial-peak pulse-average intensity, positive and negative peak pressure, sonication duration are critical for SDT treatment of cancer. Please collect important and representative literature, discuss the correlation between the parameters and treatment efficacy, and list a summary table.
  4. Line 195: please cite the references about LIPUS treatment of prostate cancer alone.
  5. Line 200 - 201: please explain why the LIPUS can penetrate deeper than HIFU.
  6. Line 203 - 220: please write the ultrasonic parameters of LIPUS for prostate cancer treatment.
  7. As mentioned in Line 192 and Line 96, LIPUS causes stable cavitation and induces microstreaming and shear stress. Is energy of microstreaming sufficiently strong to elicit sonodynamic effect? Please discuss it.
  8. For glioma treatment, please write the size of 5-ALA and fluorescein and discuss how they penetrate the BBB.
  9. Line 290: please note US is continuous or pulsed wave.
  10. For PDAC treatment, no information of SDT is described. Only HIFU is mentioned. Please add the review on SDT for PDAC treatment.
  11. Please define DDS.

Round 2

Reviewer 1 Report

No comments

Author Response

Finally, we greatly appreciate your suggestions and giving us an opportunity to improve our manuscript.

We would like to express our sincere gratitude to the referees for their insightful comments on our paper.

We feel that these comments have helped us to improve our paper significantly.

Sincerely,

Reviewer 2 Report

One more comment. Please discuss what kinds of cancers are suitable for sonodynamic therapy.

Author Response

(The authors gave the same response as above.)
